# Identification of Common Cancer Antigens Useful for Specific Immunotherapies to Colorectal Cancer and Liver Metastases

**DOI:** 10.3390/ijms26157402

**Published:** 2025-07-31

**Authors:** Jun Kataoka, Kazumasa Takenouchi, Toshihiro Suzuki, Kazunobu Ohnuki, Yuichiro Tsukada, Naoto Gotohda, Masaaki Ito, Tetsuya Nakatsura

**Affiliations:** 1Division of Cancer Immunotherapy, Exploratory Oncology Research and Clinical Trial Center, National Cancer Center, 6-5-1 Kashiwanoha, Kashiwa 277-8577, Japan; jkataoka@east.ncc.go.jp (J.K.); katakeno@east.ncc.go.jp (K.T.); toshsuzu@east.ncc.go.jp (T.S.); konuki@east.ncc.go.jp (K.O.); 2Department of Colorectal Surgery, National Cancer Center Hospital East, 6-5-1 Kashiwanoha, Kashiwa 277-8577, Japan; yutsukad@east.ncc.go.jp (Y.T.); maito@east.ncc.go.jp (M.I.); 3Course of Advanced Clinical Research of Cancer, Juntendo University Graduate School of Medicine, 2-1-1 Hongo, Bunkyo-ku, Tokyo 113-8421, Japan; ngotohda@east.ncc.go.jp; 4Department of Hepatobiliary and Pancreatic Surgery, National Cancer Center Hospital East, 6-5-1 Kashiwanoha, Kashiwa 277-8577, Japan

**Keywords:** immunohistochemical staining, human leukocyte antigen class I, cancer vaccine, chimeric antigen receptor-T (CAR-T) cell therapy, T cell receptor-T (TCR-T) cell therapy

## Abstract

Stage IV colorectal cancer has a poor prognosis, and liver metastases are prone to recurrence, even after resection. This study aimed to identify common cancer antigens, using immunohistochemical staining, as promising targets for antigen-specific immunotherapies in colorectal cancer. We analyzed expression levels and intracellular localization of seven common cancer antigens, CLDN1, EphB4, LAT1, FOXM1, HSP105α, ROBO1, and SPARC, and human leukocyte antigen (HLA) class I via immunohistochemical staining of 85 surgical specimens from primaries and liver metastases. Staining intensity and positive staining were scored to evaluate antigen expression. In 25 primaries, seven cancer antigens were expressed in 88–96% of cases, while HLA class I was expressed on the cell membrane in 80.0% of cases. In 60 liver metastases, FOXM1 and SPARC expression were approximately half that observed in the primaries. Other antigens and HLA class I were highly expressed in both. Most of the primaries and liver metastases may benefit from chimeric antigen receptor-T cell therapy targeting CLDN1, EphB4, and LAT1. Cases with high HLA class I expression may be suitable for vaccine-based and T cell receptor-T cell therapy targeting CLDN1, EphB4, LAT1, FOXM1, HSP105α, ROBO1, and SPARC for primaries and targeting antigens, excluding FOXM1 and SPARC, for liver metastases.

## 1. Introduction

Colorectal cancer (CRC) is one of the most common cancers with a rising global incidence rate [1,2,3,4]. Recent advancements in immunotherapy, including immune checkpoint inhibitors, offer promising strategies to improve outcomes in CRC [5,6,7,8,9,10]. However, stage IV CRC is associated with a poor prognosis, with a 5-year overall survival of 16.9% [11], and 80% of cases are deemed poorly resectable [2]. The liver is the most common site of distant metastasis in stage IV CRC. Although surgical resection is the cornerstone of treatment for CRC liver metastases (CRCLM), liver metastases recur in 25–30% of patients during their disease course, even after surgery [2,5,6]. This highlights the urgent need for novel therapeutic approaches to improve outcomes in this patient population. However, the role of neoadjuvant or adjuvant chemotherapy in preventing such recurrence remains controversial, with no established strategies to date.

Immune checkpoint inhibitors, CD19 chimeric antigen receptor-T (CAR-T) cells, and mRNA-based COVID-19 vaccines have demonstrated potential to mobilize robust and diverse T cell responses against cancer. Efforts have focused on developing effective cancer vaccines and CAR/T cell receptor-T cell (TCR-T) therapies. We detected the cancer-specific antigen glypican 3 (GPC3) and identified peptides that can induce CTLs in a human leukocyte antigen (HLA) A24- or A2-restricted manner. We then conducted clinical trials of GPC3 peptide vaccines for hepatocellular carcinoma, ovarian clear cell carcinoma, and pediatric cancer [12,13,14,15,16,17,18,19], where we demonstrated multiple cases of partial response in advanced cancer and recurrence prevention. Notably, seven children (male: three; female: four) with pediatric cancer who received the GPC3 vaccine have remained disease-free for approximately 10 years. Furthermore, we identified T cell receptors (TCR) from peptide-specific CTL clones derived from a pediatric hepatoblastoma survivor, which enabled the development of TCR-T cell therapy [manuscript submitted]. Moreover, we identified the cancer-specific antigen heat shock protein 105α (HSP105α), which is highly expressed in several cancers, such as CRC and esophageal cancer, and peptides capable of inducing CTLs in an HLA A24- or A2-restricted manner. We also conducted clinical trials for HSP105α peptide vaccines in patients with CRC and esophageal cancer [20,21,22]. From these trials, we verified the safety and immunological effectiveness of TCR-T cell therapy by identifying TCRs from peptide-specific CTL clones derived from patients who received the vaccine.

Cancer heterogeneity presents a challenge because targeting a single antigen is insufficient for complete eradication. Thus, we identified 10 highly immunogenic cancer-specific antigens, including GPC3 and HSP105α, which are widely expressed in solid tumors. These antigens possess peptides that can trigger many CTL responses. We predicted 72 and 73 peptides that bind to HLA-A24 and -A2 in silico from the full-length amino acid sequences of these 10 common cancer antigens. We immunized each HLA transgenic mouse with a cocktail of synthesized peptides together with the poly I:CLC three times weekly to analyze the antigen-specific immune response. As a result, 68 peptide sequences (30 and 38, respectively) were identified that had higher CTL induction ability than GPC3 298-306 and GPC3 144-152, which were used in clinical trials. Furthermore, experiments with cocktail peptide vaccines using mouse models expressing subcutaneous tumors of each antigen showed promising results in terms of safety and efficacy [23]. For antigens to be promising targets for cancer vaccines and TCR-T cell therapy, they must be expressed on cancer cells, with corresponding HLA class I expressed on the cell membrane. For CAR-T cell therapy, antigens must be expressed on the membranes of cancer cells.

Of the 10 detected antigens, AFP, GPC3, and TGFβI were not specifically expressed in CRC. In contrast, the other seven antigens were confirmed to be expressed in CRC. As membrane proteins, EphB4 (highly expressed in prostate cancer [24], breast cancer [25], head and neck cancer [26], colon cancer [27], esophagus cancer [28], skin cancer [29], pancreatic cancer [30], lung cancer [31], etc.), CLDN1 (highly expressed in ovarian cancer [32], lung cancer [33], liver cancer [34], colorectal cancer [35], oral squamous cell carcinoma [36], melanoma [37], etc.), and LAT1 (highly expressed in melanoma, lung cancer, and colon cancer [38]) were candidates. ROBO1 is highly expressed in breast cancer, pancreatic ductal adenocarcinoma [39], hepatocellular carcinoma [40], etc. Furthermore, as intracellular proteins, FOXM1 is highly expressed in breast cancer [41] and hepatocellular carcinoma [42], and SPARC is highly expressed in breast cancer [43], glioblastoma [44], melanoma, etc. [45,46]. Immunohistochemical (IHC) analysis of primary CRC and liver metastases, including cancerous and normal areas, also verified the presence of HLA class I on the cancer cell membrane. In this study, we aimed to identify common cancer antigens that are promising targets for cancer vaccines and CAR/TCR-T cell therapies for primary CRC and liver metastases.

## 2. Results

### 2.1. Expression of Seven Common Cancer Antigens and HLA Class I on Cell Membrane in 25 Cases of Primary CRC

Figure 1 shows the immunostaining patterns and scoring positive staining for each common cancer antigen and HLA class I. SPARC was additionally expressed in the cytoplasm of cancer-associated fibroblasts (CAFs) in addition to cancer cell cytoplasm, suggesting potential therapeutic relevance for targeting SPARC in both cancer cells and CAFs [47,48,49,50,51]. Although ROBO1 is a transmembrane protein receptor, its expression is predominantly cytoplasmic in primary CRC and liver metastases [52,53]. Therefore, we assessed ROBO1 expression in the cytoplasm. In normal tissues, CLDN1 was expressed in pancreatic acinar cells and skin, LAT1 in the basal layer of pharyngeal and esophageal squamous epithelium, EphB4 in vascular endothelial cells, and HSP105α in hepatocytes [54].

We included 25 patients with primary CRC in this study. The expression levels of seven common cancer antigens and HLA class I are summarized in Table 1. For the localization phenotypes of HLA class I, 20 cases (80.0%) were of the membrane phenotype, 2 cases (8.00%) were of the cytoplasmic phenotype, and 3 cases (12.0%) were of the heterogeneity phenotype combined with the cytoplasmic and membrane phenotypes. There was no significant difference in the expression of all seven common cancer antigens and HLA class I between sexes (*p* > 0.10).

### 2.2. Expression of Seven Common Cancer Antigens and HLA Class I on Cell Membrane in 60 Cases of CRC Liver Metastases

We analyzed 60 cases of CRCLM for the expression of seven common cancer antigens and HLA class I, with the ratios shown in Table 1. There was no significant difference in the expression of all seven common cancer antigens and HLA class I between sexes, consistent with the 25 cases of primary CRC (*p* > 0.10). No significant differences in cancer-specific expression levels for each cancer antigen and HLA class I in primary CRC and liver metastases were found between the naïve and chemotherapy groups (*p* > 0.40) (Appendix A). The localization phenotype of HLA class I included 37 cases (61.7%) of the membrane phenotype, 7 cases (11.7%) of the cytoplasmic phenotype, and 12 cases (28.3%) of the heterogeneity phenotype combined with the cytoplasmic and membrane phenotypes. In three cases (5.00%), HLA expression was not detected on the cytoplasm nor cell membrane.

### 2.3. Expression of Seven Common Cancer Antigens and HLA Class I on Cell Membrane in 14 Cases of Primary CRCLM

We analyzed differences in the expression patterns of seven common cancer antigens along with HLA class I among 14 cases of primary CRC with liver metastases (Table 2). These 14 cases were divided into naïve (11 cases; 78.6%) and chemotherapy (3 cases; 21.4%) groups, while the liver metastases cases were divided into naïve (5 cases; 35.7%) and chemotherapy (9 cases; 64.3%) groups.

CLDN1, EphB4, LAT1, HSP105α, and ROBO1 expression slightly decreased or remained unchanged but remained consistently high across both cancer types. FOXM1 and SPARC exhibited high expression levels in primary CRC. However, their expression levels in liver metastases were lower than those in primary cancer (Appendix A).

The high expression rate of HLA class I on cancer cell membranes was nearly equivalent in both primary CRC and liver metastases (64.2% and 78.5%, respectively). No significant difference was found in the HLA class expression score level between primary CRC and liver metastases (*p* = 0.18). In the 14 cases of primary CRC with liver metastases, the localization phenotype of HLA class I for primaries in tumor cells was as follows: 9 cases (64.2%) exhibited the membrane phenotype, 4 cases (28.5%) showed the cytoplasmic phenotype, and 1 case (7.14%) showed a heterogeneous phenotype (both cytoplasmic and membrane). In contrast, liver metastases showed 13 cases (92.9%) of the membrane phenotype and 1 case (7.14%) of the cytoplasmic phenotype. Notably, five cases (33.3%) showed a phenotypic shift from the heterogeneous to homogenous membrane phenotype, three cases (21.4%) [(b)3 cases] transitioned from the cytoplasmic to membrane phenotype, and one case (7.13%) [(a)1 case] changed from the heterogeneous to homogenous cytoplasmic phenotype.

A heatmap was created to visualize the expression levels of seven CRC-specific cancer antigens and HLA class I across each of the 14 cases described in Figure 2. The data indicate that all cases of primary CRC and CRCLM might be adaptable for CAR-T cell therapy targeting any of the three membrane-type protein cancer antigens, namely CLDN1, EphB4, and LAT1. Based on the high expression rates of HLA class I on cancer cell membranes, we assumed that cancer vaccine and TCR-T cell therapy could be effective in more than 70% of CRC or CRCLM cases that are HLA class I positive. Additionally, FOXM1, HSP105α, ROBO1, and SPARC were identified as promising therapeutic targets in CRC, while FOXM1 and SPARC were excluded in liver metastases owing to their significantly reduced expression level.

### 2.4. The Development of a Multiplex Fluorescence Immunohistochemical Staining System for Common Cancer Antigens and HLA Class I Detection

An MFIH staining system was developed to efficiently determine the expression of five common cancer antigens and HLA class I using fewer tissue sections than necessary for usual IHC staining. This approach facilitated a comprehensive assessment for personalized immunotherapy. Figure 3 shows the IHC findings of seven common cancer antigens and HLA class I for two representative cases from 14 patients with primary CRC and liver metastases. In one representative case (Case 6), HLA class I exhibited low expression levels in both primary CRC and liver metastases. Conversely, high expression levels of CLDN1 and LAT1 were observed in both primary CRC and liver metastases. MFIH analysis (Figure 4) visually confirmed the low HLA class I expression and high CLDN1 and LAT1 expression in the primary tumor. Additionally, EphB4 was prominently expressed in liver metastases alongside CLDN1 and LAT1. In another representative case (Case 9), primary CRC showed high expression levels of CLDN1, LAT1, FOXM1, HSP105α, and SPARC, while FOXM1 expression was markedly reduced in liver metastases. MFIH analysis of Case 9 also validated these findings, clearly demonstrating the decreased FOXM1 expression in liver metastases compared to that in primary CRC.

## 3. Discussion

Cancer vaccines, such as peptide and mRNA vaccines, elicit anti-cancer effects by inducing CTLs against tumor cells expressing specific cancer antigens. Our experiments with cocktail peptide vaccines using HLA transgenic mice models expressing subcutaneous tumors of each antigen showed promising results in terms of safety and efficacy. The peptides identified in this study, derived from 10 common cancer antigens covering all solid cancers, are expected to be clinically applicable as cocktail peptide vaccines [23]. T cell therapies, including CAR-T and TCR-T cell therapies, have also gained prominence [55,56,57]. The critical role of HLA class I in these therapies lies in their function as mediators of CTL recognition and activation, making them essential for cancer vaccines and TCR-T cell therapy. Conversely, cancer cells frequently evade immune surveillance through partial or total loss of HLA class I expression, thus reducing their susceptibility to CTL-mediated clearance [58,59,60].

The previously reported frequencies of HLA class I between primary CRC and liver metastases (72–74% [61,62,63] and 64–66.4% [62,64]) align closely with those of our study (61.7% and 78.6%). Meanwhile, the observed difference in HLA class I phenotypes between primary CRC and liver metastases in the same patients may result from clonal heterogeneity within the primary tumor. Some metastatic clones may retain the HLA class I phenotype of their primary tumor of origin, while others may acquire new mutations, leading to altered phenotypes [65]. Another possibility is the metastatic spread of HLA class I-positive tumor cells from heterogeneous primary tumors, as observed in liver metastasis derived from mismatch repair-deficient/high-frequency microsatellite instability CRC [66]. High HLA class I expression in such cases has been associated with poor prognosis [67]. Furthermore, HLA class I-expressing tumor cells remain, even after HLA class I-negative tumor nests are eradicated by NK cells. These surviving cells can seed distant organs, especially the liver, and form metastases [65].

Regarding cancer antigens, our findings highlight the expression of FOXM1 and SPARC as being significantly lower in liver metastases than in primary CRC. Both FOXM1 and SPARC, almost undetectable in normal liver tissues [47,68], facilitate cancer cell migration and invasion in CRC [48,69]. The role of FOXM1 is limited to driving the migration and invasion of cancer cells to distant organs, and its expression may not be significantly expressed or reflected in liver metastases. Similarly, SPARC, although contributing to the invasive and metastatic properties of cancer cells [47], may also exhibit reduced expression in liver metastases due to the minimal stromal content in liver tissue. In addition, the decreased expression of FOXM1 and SPARC in liver metastases compared to primary tumors may mainly reflect the phenotypic changes associated with metastatic progression rather than other cancer antigens. These characteristics likely explain the significant reduction in FOXM1 and SPARC expression and reflection in liver tissue compared to the primaries. Other cancer antigens, while exhibiting downregulated expression in liver tissue compared to primary tumors, may not exhibit as marked a decline as FOXM1 and SPARC, underscoring their distinct biological roles in tumor progression and metastasis.

Our study revealed no significant differences in HLA class I expression between primary CRC and liver metastases. Approximately 70% of patients demonstrated HLA class I expression localized to cancer cell membranes in both primary and liver metastases. Comparable results have been reported by Michelakos T et al. in 2022 [70]. Regarding the deficiency of HLA-ABC in CRC, they performed a comprehensive analysis with five large cohort studies of CRC and reported that the percentage of CRC with total HLA (HLA-ABC) deficiency was 16%, and 84% of CRC cases expressed either HLA-A, B, or C. Using whole exome sequencing (WES) analysis, Li C et al. also reported the positive incidence of LOH in HLA locus in primary CRC and CRCLM [71]. Although there was no significance, the frequency of LOH-HLA in CRCLM showed a tendency to increase compared with that in primary CRC: the positive incidence of LOH HLA was 26% (4/15) in all patients, 13% (2/15) in primary CRC, and 26% (4/15) in liver metastases (CRCLM), respectively [71]. Meanwhile, in our house data, 31.8% of patients with CRCLM showed putative loss of heterozygosity of the HLA gene. Compared with immunohistochemically analysis with RPM8-5, we found that some cases with LOH of HLA-A, B, or C showed positive staining of HLA-ABC, since staining with RPM8-5 could not clearly distinguish cases with LOH in only one HLA gene locus, and the loss of either HLA-A, B, or C depends on its staining intensity.

CLDN1, EphB4, LAT1, FOXM1, HSP105α, ROBO1, and SPARC were identified as potential targets for common cancer antigens for primary CRC. However, FOXM1 and SPARC may have limited applicability as therapeutic targets for liver metastases. Membrane proteins, such as CLDN1, EphB4, and LAT1, were highly expressed in primary CRC and liver metastases, suggesting that CAR-T cell therapy targeting these three cancer antigens could be a viable option, particularly in 30% of cases with reduced HLA class I expression on cancer cell membranes.

Our study has some limitations. Analyses were performed at a single center, which may have limited the generalizability of the findings because fewer samples were used. As our immunohistochemical analysis represents a snapshot of antigen expression in tissue sections, it may not fully reflect the heterogeneity of cancer antigen and HLA class I in the entire tumor. The MFIH system requires further modification, including improvements to scoring methods. Of the 60 CRCLM cases analyzed, no significant differences were observed in the expression of cancer antigens or HLA class I between 12 naïve and 48 chemotherapy cases. However, typically, the specimens of liver metastases were not uniformly distributed owing to various background factors, such as liver recurrence after resection of primary CRC or liver metastases of CRC following preoperative treatment. Therefore, this heterogeneity in the background of liver metastasis cases, although not significantly affecting our results, should be considered when interpreting these findings. Although this study compared 25 cases of primary CRC with 60 cases of liver metastases, the lack of identical case pair comparisons remains a limitation. Meanwhile, in terms of HLA class I, our studies used EMR8-5, a novel pan-HLA monoclonal antibody, which is suitable for HLA class I immunostaining in formalin-fixed paraffin-embedded tissue specimens. The EMR8-5 antibody recognizes all HLA-A, -B, and -C heavy chains, even in formalin-fixed tissues, and these results could estimate whether cells are recognized by CTLs [72].

Despite this, we observed that the median expression levels of all seven cancer antigens and HLA class I were lower in liver metastases than those in primary CRC. Further analysis of 14 identical cases of primary CRC and liver metastases (Appendix A) showed that while the median expression levels of FOXM1 and SPARC were significantly decreased in liver metastases, the median expression levels of CLDN1, EphB4, LAT1, HSP105α, ROBO1, and HLA class I were only slightly reduced and not statistically significant. Importantly, the median expression levels of CLDN1, EphB4, LAT1, HSP105α, ROBO1, and HLA class I remained above 3.5, and their frequencies exceeded 50% in liver metastases. These findings suggest that CLDN1, EphB4, LAT1, HSP105α, and ROBO1 may serve as promising targets for immunotherapy in liver metastases. The identification of common cancer antigens adaptable for vaccine development and CAR/TCR-T cell therapy holds great value for advancing immunotherapeutic strategies. One major obstacle is tumor heterogeneity in CRC tumors. CRC cells can exhibit significant genetic and phenotypic variability, leading to the expression of different antigens within the same tumor or between primary and metastatic sites [73,74]. This heterogeneity makes it difficult for CAR-T cells, which were traditionally engineered to target a single antigen, to effectively eliminate all cancerous cells. The risk of antigen loss or mutation can result in tumor escape variants, where cancer cells that no longer express the target antigen adapt and spread, causing a relapse. Moreover, in the hostile environment of CRC, CAR-T cells often exhibit reduced persistence and activity, limiting their therapeutic potential. Therefore, it is necessary to create dual-targeting or logic-gated CAR designs. Clinical trials using CEA-targeted CAR-T cells have demonstrated promising disease stabilization, although the risk of on-target, off-tumor toxicity remains due to CEA expression in normal gastrointestinal epithelium [75]. Accordingly, multi-antigen targeting approaches, including tandem CARs or AND-gated systems (e.g., CEA + HLA-A02 LOH), may offer improved tumor specificity and reduced toxicity risk [76].

These therapeutic strategies hold promise for treating primary CRC and liver metastases. Based on our findings, further in vivo and in vitro studies are planned to validate the potential of these identified antigens as targets for immunotherapy in CRC liver metastases (e.g., EphB4 CAR-T trial for EphB4-positive CRCLM).

## 4. Materials and Methods

### 4.1. Clinical Samples

We included surgical cases based on two criteria: (1) recent cases with well-preserved surgical specimen blocks and (2) cases where both tumor and normal tissues were preserved within the same block to allow comparative analyses. We obtained 85 surgical specimens from 25 patients (male: 15; female: 10) with primary CRC (including 14 primary CRC cases with concurrent liver metastases) and 60 patients (male: 42; female: 18) with CRCLM who underwent surgery at our hospital between December 2016 and July 2021. There was no difference in surgical outcomes and prognoses between the sexes. The CRCLM cases included in our study were categorized into two groups: (a) the naïve group, including patients who had not received chemotherapy and underwent simultaneous resection of the primary tumor or liver metastases alone, and (b) the chemotherapy group, including patients who received chemotherapy either before or after primary CRC resection.

### 4.2. Immunohistochemical Analysis of Common Cancer Antigens and HLA Class I

For IHC analysis, we used specific antibodies against the following seven common cancer antigens and HLA class I: Claudin 1 [CLDN1 (#13255; Cell Signaling Technology^®^ Inc., Danvers, MA, USA, 300-fold dilution)], Ephrin type-B receptor 4 [EphB4 (#14960; Cell Signaling Technology^®^, Inc., Danvers, MA, USA, 300-fold dilution)], L-type amino acid transporter 1 [LAT1 (ab208776; Abcam^®^, Cambridge, UK, 200-fold dilution)], Forkhead box M1 [FOXM1 (ab207298; Abcam^®^, Cambridge, UK, 150-fold dilution], heat shock protein 105α [HSP105α (MA5-32408; Novus Bio ^®^, Centennial, CO, USA, 200-fold dilution)], roundabout homolog-1 [ROBO1 (25181-1-AP; Proteintech^®^, Rosemont, IL, USA, 300-fold dilution)], secreted protein acidic and rich in cysteine [SPARC (sc-73472; Santa Cruz Biotechnology^®^, Dallas, TX, USA, 250-fold dilution], and HLA class I [HLA-ABC (IR500; HOKUDO^®^, Sapporo, Japan, 200-fold dilution)]. Formalin-fixed paraffin-embedded tissues (4 μm) were deparaffinized with xylene and rehydrated with ethanol. Endogenous peroxidase activity was blocked using 0.3% H_2_O_2_/MeOH. Antigen activation was performed via microwave heat treatment in TRS9 buffer (pH 9.0, Dako/Agilent, Santa Clara, CA, USA) and AR6 buffer (pH 6.0, Akoya Biosciences, Marlborough, MA, USA) at 95 °C or 121 °C for 20 min. Primary antibodies were incubated at room temperature (20–25° C) for 1 h or overnight at 4 °C. Mouse/rabbit Envision Polymer (Dako/Agilent, Santa Clara, CA, USA) was used as the secondary antibody and incubated at room temperature for 30 min, followed by 3, 3′-diaminobentidine staining and hematoxylin counterstaining. Slides were dehydrated with xylene and sealed with glass coverslips.

Staining intensity of the tumor tissue was scored for each of the seven antigens using a semi-quantitative method (negative: 0, weakly positive: 1, weakly to strongly positive: 2, and strongly positive: 3), while the percentages of positively stained cell areas were scored as follows: 0 (0–10%), 1 (10–39%), 2 (40–69%), and 3 (70–100%). We defined the sum of these two scores as the total score.

Total Score (0~6) =

Average scores of staining intensities from three regions of interest (ROIs): 0~3 

+ Average scores of positive areas from three ROIs: 0~3.

The individual scores were averaged across three random visual fields (40x field of view). Intracellular localization of tumor tissue (membrane, cytoplasm, or nucleus) was also evaluated using IHC analysis [77]. Virtual slides were created using Nano Zoomer (Hamamatsu Photonics), and three researchers independently analyzed staining using these virtual slides at a 40x field of view. A total score of 3.1–6.0 based on IHC analyses was defined as high expression.

### 4.3. Multiplex Fluorescence Immunohistochemical Staining and Analysis of Cancer Antigens and HLA Class I

Multiplex fluorescent immunohistochemical staining (MFIH) was performed on 4 μm thick tissue sections of primary CRC and liver metastases using the PerkinElmer Opal kit. MFIH images were acquired using the tissue section quantitative analysis imaging system (Vectra 3,; Akoya Biosciences, Marlborough, MA, USA). Up to 20 ROIs (699 × 500 µm) were randomly selected and evaluated in each tumor center and margins using an image analysis program (Inform 2.6.; PerkinElmer, Inc., Shelton, CT, USA). The distribution of specific cancer antigens was also analyzed. ROBO1 and SPARC were excluded owing to their complex intracellular localization, with ROBO1 being expressed in the nucleus, membrane, and cytoplasm of cancer cells and SPARC in both the cytoplasm of cancer cells and CAFs. Based on these characteristics, the MFIH panel was finalized with CLDN1, EphB4, LAT1, FOXM1, HSP105α, and HLA class I.

### 4.4. Statistical Analysis

Patient characteristics were summarized using descriptive statistics, such as medians (range) and proportions. Univariate analyses were conducted to assess differences between primary CRC and liver metastases using the Mann–Whitney U test for continuous variables and Fisher’s exact test for categorical variables. Statistical significance was set at *p* < 0.05. We also created a heatmap to depict the expression levels of each cancer antigen and HLA class I in the 14 cases of primary CRC with liver metastases. All statistical analyses were performed using EZR software (version 1.6.6, Jichi Medical University, Tochigi, Japan) [78]) for R (version 4.2.2, The R Foundation for Statistical Computing, Vienna, Austria).

## 5. Conclusions

Our study highlights that most cases of primary CRC could potentially benefit from CAR-T cell therapy targeting CLDN1, EphB4, and LAT1. Furthermore, CRC with high levels of HLA class I expression on the cell membrane may be suitable for vaccine-based and TCR-T cell therapies targeting CLDN1, EphB4, LAT1, FOXM1, HSP105α, ROBO1, and SPARC. For liver metastases, while CAR-T cell therapy targeting CLDN1, EphB4, and LAT1 remains a viable option, vaccines and TCR-T cell therapy targeting CLDN1, EphB4, LAT1, HSP105α, and ROBO1—excluding FOXM1 and SPARC—may be effective in most cases. These findings underscore the potential for personalized immunotherapeutic approaches tailored to the antigenic profiles of primary and metastatic CRCs.

## Figures and Tables

**Figure 1 ijms-26-07402-f001:**
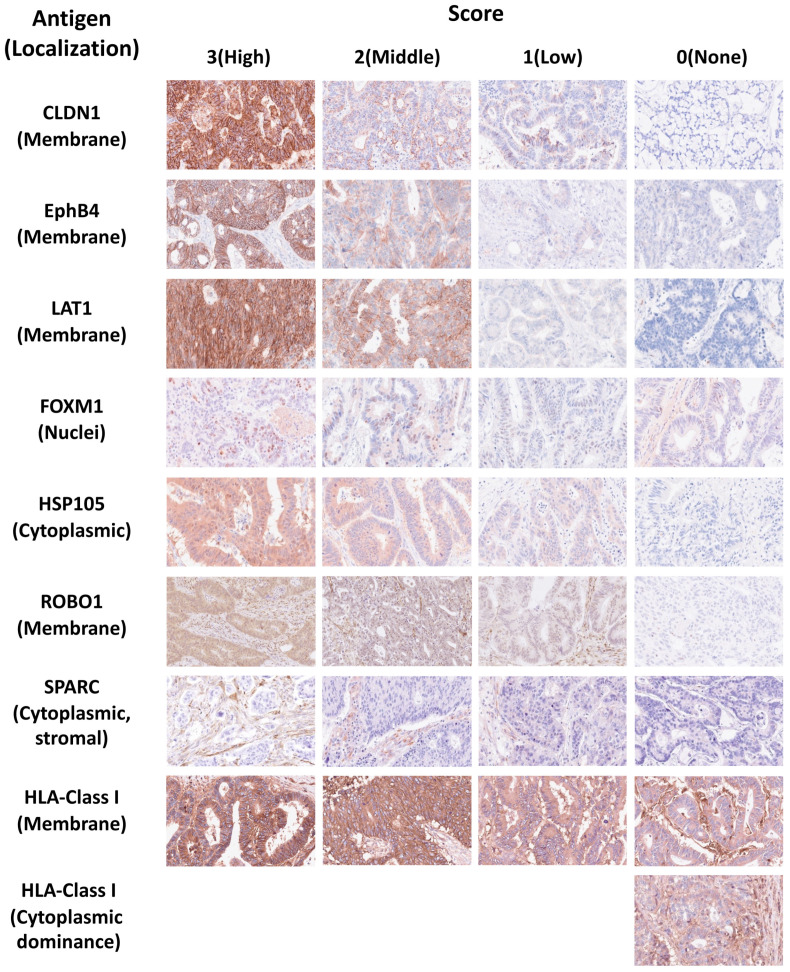
The scoring of immunohistochemical staining intensity for common cancer antigens and phenotypes and the scoring of staining intensity for HLA class I. The staining intensity of cancer cells for each of the eight antigens—CLDN1, EphB4, LAT1, FOXM1, HSP105α, ROBO1, SPARC, and HLA class I—was scored as follows: negative: 0, weakly positive: 1, weakly to strongly positive: 2, and strongly positive: 3. Intracellular localization of tumor tissue was evaluated using immunohistochemical analysis. CLDN1, EphB4, and LAT1 were mostly expressed on the cell membrane of cancer cells, FOXM1 was expressed on the cell nuclei, HSP105α and ROBO1 were expressed in the cytoplasm of cancer cells, and SPARC was expressed in the cytoplasm of cancer-associated fibroblasts (CAFs) [31]. For HLA class I expression, the presence and staining of HLA class I were observed in both the cell membrane and cytoplasm, but only staining on the membranes of cancer cells was evaluated.

**Figure 2 ijms-26-07402-f002:**
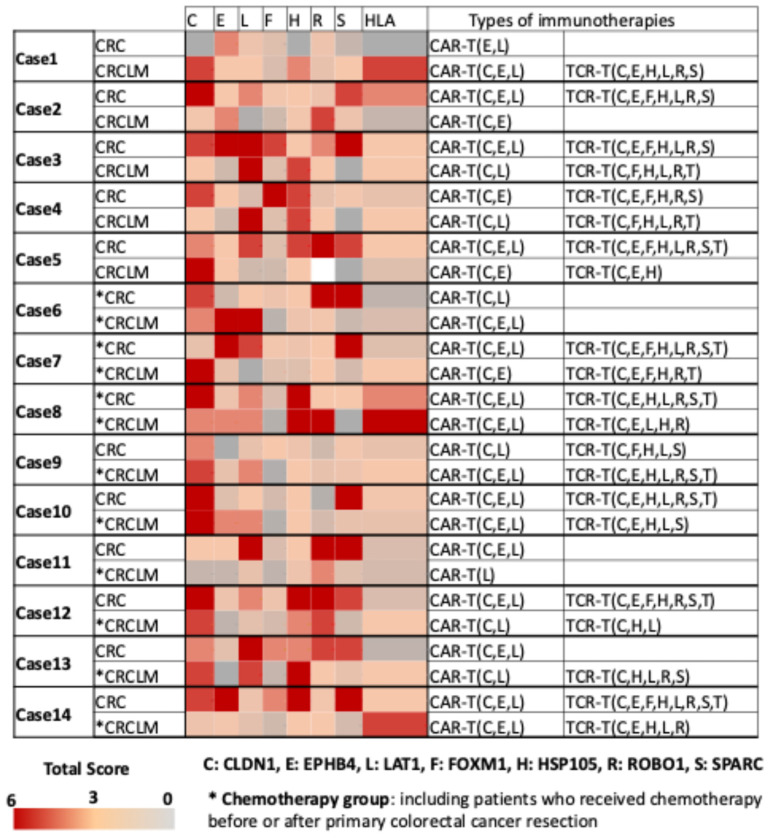
Heatmap and expression score table of seven common cancer antigens and HLA class I in 14 cases of primary colorectal cancer and liver metastases. The median core values of staining intensity and positive staining for seven common cancer antigens—CLDN1, EphB4, LAT1, FOXM1, HSP105α, ROBO1, and SPARC—and HLA class I on tumor cells from 14 cases each of primary colorectal cancer and liver metastases. The expression levels of each antigen were evaluated, alongside potential T cell therapies (CAR/TCR-T cell therapy) applicable to each case. High expression was defined as a score of >3. Cases with high HLA class I expression were suitable for TCR-T, while those with low HLA class I expression may benefit from CAR-T targeting highly expressed CLDN1, EphB4, and LAT1. The cases were divided into naïve and chemotherapy groups. * The chemotherapy group, including patients who received chemotherapy before or after primary colorectal cancer resection. C: CLDN1, E: EphB4, L: LAT1, F: FOXM1, H: HSP105α, R: ROBO1, S: SPARC, HLA: HLA class I, CAR-T: chimeric antigen receptor-T, TCR-T: T cell receptor-T.

**Figure 3 ijms-26-07402-f003:**
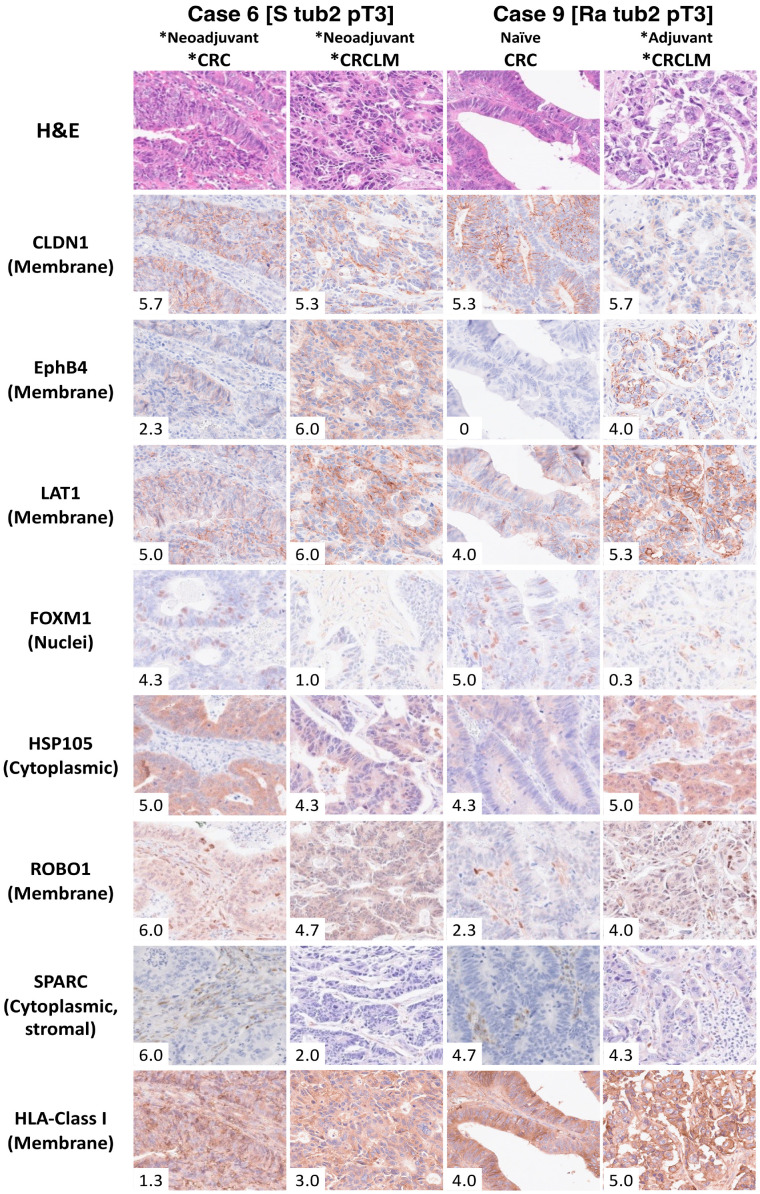
Immunohistochemical staining and analysis of seven common cancer antigens. Two representative cases of 14 patients with primary colorectal cancer and liver metastases are presented. The total expression scores for seven cancer antigens and HLA class I in these two representative cases are displayed in the lower-left corner of each image. Case 6, the patient who received chemotherapy prior to primary and liver metastases resection, exhibited high expression of FOXM1, HSP105α, ROBO1, and SPARC in the primary tumor, whereas HLA class I expression was low. CLDN1 and LAT1 were highly expressed, suggesting potential suitability for CAR-T therapy targeting CLDN1 and LAT1. For the liver metastases of Case 6, most antigens showed high expression, except FOXM1 and SPARC, while HLA class I expression remained low. Case 9 represents the patient who underwent primary tumor resection without prior chemotherapy but received chemotherapy before liver metastasis resection. In this case, the primary tumor showed high expression of HLA class I, CLDN1, LAT1, FOXM1, HSP105α, and SPARC. Hence, this case was deemed suitable for TCR-T therapy targeting CLDN1, LAT1, FOXM1, HSP105α, and SPARC and for CAR-T cell therapy targeting CLDN1 and LAT1. The liver metastases of Case 9 displayed consistent HLA class I expression, but FOXM1 expression was notably decreased compared with the primary tumor, suggesting that TCR-T cell therapy targeting antigens excluding FOXM1, along with CAR-T cell therapy targeting CLDN1, EphB4, and LAT1, may be applicable. * The chemotherapy group, including patients who received chemotherapy before primary CRC or CRCLM resection.

**Figure 4 ijms-26-07402-f004:**
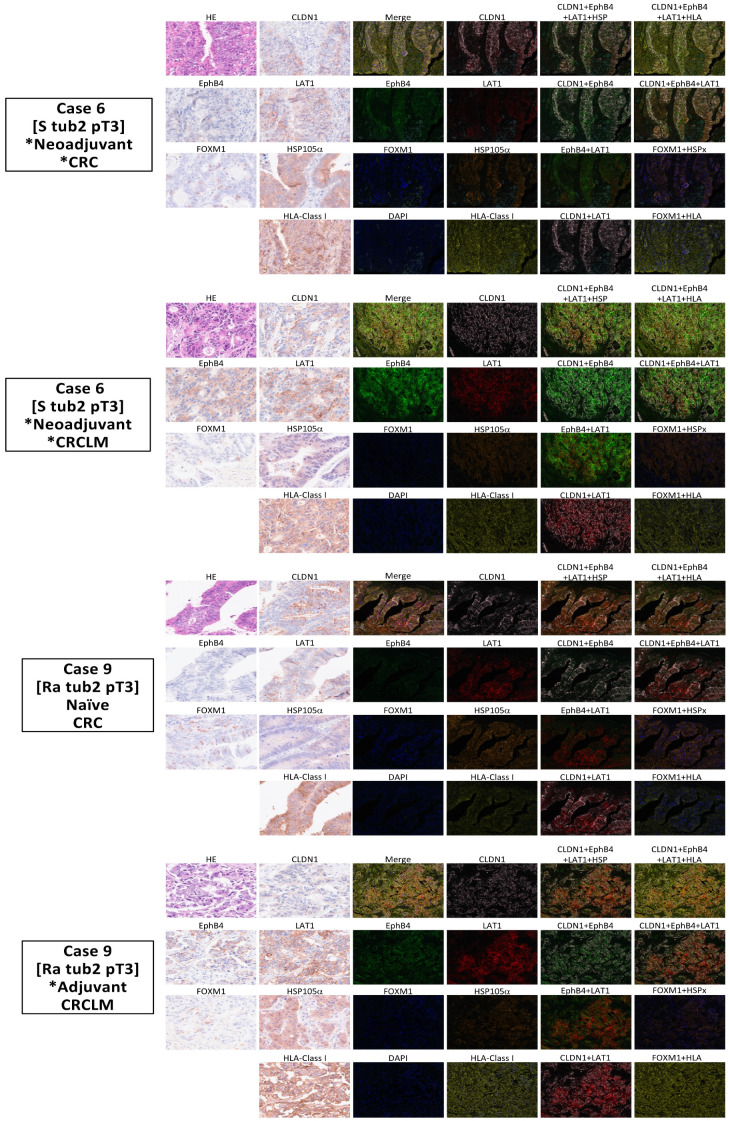
Multiplex fluorescence immunohistochemical (MFIH) staining and analysis of five common cancer antigens and HLA class I. MFIH staining was carried out using specific antibodies: opal 520 (green) for anti-EphB4, opal 540 (blue) for anti-FOXM1, opal 570 (red) for anti-LAT1, opal 620 (orange) for anti-HSP105α, opal 650 (yellow) for anti-HLA class I, and opal 690 (pink) for anti-CLDN1. MFIH results for Case 6 showed low expression of HLA class I in both primary colorectal cancer and liver metastases. CLDN1 and LAT1 were visually confirmed to be highly expressed in primary tumors, with EphB4 also showing high expression in liver metastases along with CLDN1 and LAT1. MFIH analysis of Case 9 visually confirmed uniformly high expression of five cancer antigens and HLA class I in primary tumor. However, FOXM1 expression was reduced in liver metastases. * The chemotherapy group, including patients who received chemotherapy before primary CRC or CRCLM resection.

**Table 1 ijms-26-07402-t001:** Expression of seven kinds of common cancer antigens and expression of HLA class I on cell membrane in 25 cases of primary colorectal cancer and 60 cases of colorectal cancer liver metastases.

Primary Colorectal Cancer								
Cancer Antigens *	CLDN1	EphB4	LAT1	FOXM1	HSP105α	ROBO1	SPARC	HLA class I
Score: Median [range]								
All cases (n = 25)	5.3 [0–6.0]	5.0 [0–6.0]	5.3 [0.7–6.0]	5.0 [0–6.0]	6.0 [3.0–6.0]	4.7 [0.7–6.0]	5.7 [1.7–6.0]	4.7 [0–5.3]
Cases of expression > 3								
All cases (n = 25)	23 (92.0%)	24 (96.0%)	24 (96.0%)	22 (88.0%)	24 (96.0%)	23 (92.0%)	24 (96.0%)	20 (80.0%)
Colorectal cancer liver metastases								
Cancer Antigens *	CLDN1	EphB4	LAT1	FOXM1	HSP105α	ROBO1	SPARC	HLA class I
Score: Median [range]								
All cases (n = 60)	4.3 [0–6.0]	4.3 [0–6.0]	3.7 [0–6.0]	2.7 [0–6.0]	5.0 [0–6.0]	4.0 [0–6.0]	2.7 [0–5.0]	3.7 [0–6.0]
Naïve (n = 12)	4.3 [3.3–6.0]	5.0 [0–6.0]	3.7 [1.3–5.7]	2.5 [0–5.0]	4.7 [3.3–6.0]	4.2 [0–6.0]	2.3 [1.0–5.0]	3.7 [0–5.0]
Chemotherapy (n = 48)	4.3 [0–6.0]	4.0 [0–6.0]	3.7 [0–6.0]	2.7 [0–6.0]	5.0 [0–6.0]	3.7 [0–6.0]	2.7 [1.0–4.0]	3.7 [0–6.0]
Cases of expression > 3								
All cases (n = 60)	53 (88.3%)	44 (76.6%)	36 (60.0%)	23 (38.3%)	57 (95.0%)	38 (63.3%)	20 (33.3%)	37 (61.7%)
Naïve (n = 12)	12 (100.0%)	8 (66.7%)	7 (58.3%)	2 (16.7%)	12 (100.0%)	8 (66.7%)	2 (16.7%)	10 (83.3%)
Chemotherapy (n = 48)	41 (85.4%)	36 (75.0%)	29 (60.4%)	21 (43.8%)	45 (93.8%)	30 (62.5%)	18 (37.5%)	27 (56.2%)

* CLDN1: Claudin 1, EphB4: Ephrin Type-B Receptor 4, LAT1: L-Type Amino Acid Transporter 1, FOXM1: Forkhead Box M1, HSP105α: Heat Shock Protein 105α, ROBO1: Roundabout Homolog-1, SPARC: Secreted Protein Acidic and Rich in Cysteine.

**Table 2 ijms-26-07402-t002:** Expression of seven kinds of common cancer antigens and HLA class I on cell membrane in 14 patients of primary colorectal cancer with liver metastases.

Primary Colorectal Cancer								
Cancer Antigens *	CLDN1	EphB4	LAT1	FOXM1	HSP105α	ROBO1	SPARC	HLA class I
Score: Median [range]								
All cases (n = 14)	5.7 [0–6.0]	5.0 [0–6.0]	5.3 [3.0–6.0]	4.5 [2.7–6.0]	5.0 [0–6.0]	5.0 [0.7–6.0]	5.7 [1.7–6.0]	4.0 [0–5.3]
Naïve (n = 11)	5.7 [0–6.0]	5.0 [5.0–6.0]	5.3 [3.0–6.0]	4.7 [2.7–6.0]	5.0 [0–6.0]	4.7 [0.7–6.0]	5.7 [1.7–6.0]	4.0 [0–5.3]
Chemotherapy (n = 3)	5.7 [4.0–6.0]	4.3 [2.3–6.0]	5.3 [5.0–5.7]	4.3 [3.0–4.7]	5.20 [4.3–6.0]	5.0 [5.0–6.0]	6.0 [5.0–6.0]	3.3 [1.3–5.3]
Cases of expression > 3								
All cases (n = 14)	13 (92.9%)	12 (85.7%)	13 (92.9%)	11 (78.6%)	13 (92.9%)	12 (85.7%)	13 (92.9%)	9 (64.2%)
Naïve (n = 11)	10 (90.9%)	10 (90.9%)	10 (90.9%)	9 (81.8%)	10 (90.9%)	9 (81.8%)	10 (90.9%)	7 (63.6%)
Chemotherapy (n = 3)	3 (100.0%)	2 (66.7%)	3 (100.0%)	2 (66.7%)	3 (100.0%)	3 (100.0%)	3 (100.0%)	2 (66.7%)
Colorectal cancer liver metastases								
Cancer Antigens *	CLDN1	EphB4	LAT1	FOXM1	HSP105α	ROBO1	SPARC	HLA class I
Score: Median [range]								
All cases (n = 14)	5.5 [1.7–6.0]	4.3 [0–6.0]	5.2 [0–6.0]	2.3 [0–3.3]	5.0 [3.3–6.0]	5.0 [4.0–6.0]	2.7 [0–5.0]	5.0 [1.7–6.0]
Naïve (n = 5)	5.0 [1.7–5.7]	5.0 [2.3–5.3]	5.0 [0.3–6.0]	3.0 [2.3–3.3]	5.3 [4.7–5.7]	5.0 [4.0–5.7]	0.0 [0–5.0]	5.0 [1.7–5.7]
Chemotherapy (n = 9)	5.0 [1.7–6.0]	4.0 [0–6.0]	5.3 [0–6.0]	1.3 [0.3–3.3]	5.0 [3.3–6.0]	5.0 [2.7–6.0]	2.7 [0–4.3]	5.0 [2.7–5.7]
Cases of expression > 3								
All cases (n = 14)	13 (92.9%)	9 (64.2%)	11 (78.6%)	4 (28.6%)	14 (100.0%)	12 (85.7%)	5 (35.7%)	11 (78.6%)
Naïve (n = 5)	5 (100.0%)	3 (60.0%)	3 (60.0%)	3 (60.0%)	5 (100.0%)	4 (80.0%)	2 (40.0%)	4 (80.0%)
Chemotherapy (n = 9)	8 (88.9%)	6 (66.7%)	8 (88.9%)	1 (11.1%)	9 (100.0%)	8 (88.9%)	3 (33.3%)	7 (77.8%)

* CLDN1: Claudin 1, EphB4: Ephrin Type-B Receptor 4, LAT1: L-Type Amino Acid Transporter 1, FOXM1: Forkhead Box M1, HSP105α: Heat Shock Protein 105α, ROBO1: Roundabout Homolog-1, SPARC: Secreted Protein Acidic and Rich in Cysteine.

## Data Availability

The data presented in this study are available on request from the corresponding author.

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
