# Peer review of "Identification of Common Cancer Antigens Useful for Specific Immunotherapies to Colorectal Cancer and Liver Metastases"

_ijms, 2025, doi:10.3390/ijms26157402_

Round 1

Reviewer 1 Report

Comments and Suggestions for Authors

This manuscript investigates the identification of novel tumor antigens suitable for the development of CAR-T cell immunotherapy targeting metastatic colorectal cancer. Additionally, the authors assess both primary colorectal carcinomas (CRCs) and liver metastases for HLA class I expression to evaluate the feasibility of TCR-T cell therapy in tumors retaining positive HLA-I expression. The topic is timely and significant, especially given the growing interest in enhancing CAR-T cell efficacy against solid tumors. The study's primary findings are based on the histological distribution of antigens in tissue samples from primary colorectal tumors and liver metastases. However, the immunohistochemistry (IHC) images presented are of such inadequate quality that it is difficult to discern the labeling patterns referenced by the authors, hindering the interpretation and discussion of the results. Furthermore, several of the antigens analyzed are not commonly associated with colorectal cancer and in the text the relevant information is moslty supported by self references of the authors.  These antigens should be more thoroughly described in the Introduction and Discussion sections, including details on their common cellular and subcellular localization and their expression across different tissues and cancer types, particularly in the context of immunotherapeutic application. Moreover, the authors should expand the Discussion to acknowledge the well-documented challenges associated with CAR-T therapy in solid tumors. These include antigenic heterogeneity and the immunosuppressive tumor microenvironmet that prevent tumor infoltration with CAR-Ts.

The authors should exercise caution in interpreting the HLA class I labeling results. It is well established that accurate assessment of cell surface expression of the functional HLA class I complex (heavy chain, β2-microglobulin, and tumor-derived peptide) requires the use of fresh-frozen tissue sections and detection with antibodies such as W6/32, which specifically recognize the intact functional complex. In formalin-fixed, paraffin-embedded (FFPE) tissues, this complex is not preserved. While the alternative antibody employed in this study is capable of detecting free HLA-A and -B heavy chains in FFPE samples, positive membrane staining alone does not confirm the presence of the complete, functional HLA class I complex on tumor cells, which is relevant for the purpose of this study - search for TCR-T cell therapy for HLA-I positive tumors. This limitation should be clearly acknowledged in the interpretation of the findings. Furthermore, the reported frequency of HLA class I loss in this study appears considerably lower than what has been documented by other groups. This discrepancy should be discussed, as it may reflect earlier described methodological details, such as tissue preservation and antibody specificity.

Author Response

For research article

Identification of common cancer antigens useful for specific immunotherapies to colorectal cancer and liver metastases

Response to Reviewer 1 Comments

1. Summary

Thank you very much for reviewing and commenting on our manuscript. We have revised our manuscript in accordance with your comments and suggestions. The revised parts are shown in red font in the revised version of the manuscript. Our point-by-point responses to the reviewer`s comments are given below, and we have made every effort to address the issues raised and respond to all the comments. We hope that our revisions will meet the reviewers’ expectations. Kindly refer to the page and line numbers in the revised version of the manuscript.

2. Questions for General Evaluation

Reviewer’s Evaluation

Response and Revisions

Does the introduction provide sufficient background and include all relevant references?

Can be improved

Are all the cited references relevant to the research?

Yes

Is the research design appropriate?

Yes

Are the methods adequately described?

Can be improved

Are the results clearly presented?

Can be improved

Are the conclusions supported by the results?

Must be improved

3. Point-by-point response to Comments and Suggestions for Authors

This manuscript investigates the identification of novel tumor antigens suitable for the development of CAR-T cell immunotherapy targeting metastatic colorectal cancer. Additionally, the authors assess both primary colorectal carcinomas (CRCs) and liver metastases for HLA class I expression to evaluate the feasibility of TCR-T cell therapy in tumors retaining positive HLA-I expression. The topic is timely and significant, especially given the growing interest in enhancing CAR-T cell efficacy against solid tumors. The study's primary findings are based on the histological distribution of antigens in tissue samples from primary colorectal tumors and liver metastases.

(Point.1)

However, the immunohistochemistry (IHC) images presented are of such inadequate quality that it is difficult to discern the labeling patterns referenced by the authors, hindering the interpretation and discussion of the results. Furthermore, several of the antigens analyzed are not commonly associated with colorectal cancer and in the text the relevant information is mostly supported by self references of the authors.

(Point.2)

These antigens should be more thoroughly described in the Introduction and Discussion sections, including details on their common cellular and subcellular localization and their expression across different tissues and cancer types, particularly in the context of immunotherapeutic application.

(Point.3)

Moreover, the authors should expand the Discussion to acknowledge the well-documented challenges associated with CAR-T therapy in solid tumors. These include antigenic heterogeneity and the immunosuppressive tumor microenvironment that prevent tumor infiltration with CAR-Ts.

(Point.4)

The authors should exercise caution in interpreting the HLA class I labeling results. It is well established that accurate assessment of cell surface expression of the functional HLA class I complex (heavy chain, β2-microglobulin, and tumor-derived peptide) requires the use of fresh-frozen tissue sections and detection with antibodies such as W6/32, which specifically recognize the intact functional complex.

(Point.5)

In formalin-fixed, paraffin-embedded (FFPE) tissues, this complex is not preserved. While the alternative antibody employed in this study is capable of detecting free HLA-A and -B heavy chains in FFPE samples, positive membrane staining alone does not confirm the presence of the complete, functional HLA class I complex on tumor cells, which is relevant for the purpose of this study - search for TCR-T cell therapy for HLA-I positive tumors. This limitation should be clearly acknowledged in the interpretation of the findings. Furthermore, the reported frequency of HLA class I loss in this study appears considerably lower than what has been documented by other groups. This discrepancy should be discussed, as it may reflect earlier described methodological details, such as tissue preservation and antibody specificity.

Response 1: (Answer for point.1)

Thank you very much for reviewing and commenting on our manuscript. We revised the quality of figure more clearly according to your suggestion. Please confirm these figures.(Page 5,11, and 13)

Response 2: (Answer for point.2)

Thank you for important points to improve our manuscript. These points will be stated and added in our manuscript, as below.

(Page 2 lines 83-92, as the statements on the expression of each cancer antigen in vary cancers)

As membrane proteins, EphB4: highly expressed in prostate cancer [24], breast cancer [25], head and neck cancer [26], colon cancer [27], esophagus cancer [28], skin cancer [29], pancreatic cancer [30], lung cancer [31], etc. CLDN1: highly expressed in ovarian cancer [32], lung cancer [33], liver cancer [34], colorectal cancer [35], oral squamous cell carcinoma [36], melanoma [37], etc. LAT1: highly expressed in melanoma, lung cancer, and colon cancer [38], were candidates. ROBO1: highly expressed in breast cancer, pancreatic ductal adenocarcinoma [39], hepatocellular carcinoma [40], etc. Furthermore, as intracellular proteins, FOXM1: highly expressed in breast cancer [41] and hepatocellular carcinoma [42], SPARC: highly expressed in breast cancer [43], glioblastoma [44], melanoma, etc. [45,46].

(Page 5 lines 153-158, Figure 1 legend, as the statement on the localization of cancer antigens)

 CLDN1, EphB4, and LAT1 were mostly expressed on the cell membrane of cancer cells, FOXM1 was expressed on the cell nuclei, HSP105α and ROBO1 were expressed in the cytoplasm of cancer cells, and SPARC was expressed in the cytoplasm of cancer-associated fibroblasts (CAF).

Response 3: (Answer for point.3)

We have revised to your questions about CAR-T therapy in solid tumors, and added the following to our manuscript.

(Page 10 lines 315-329)

One of major obstacles is tumor heterogeneity in CRC tumors. CRC cells can exhibit significant genetic and phenotypic variability, leading to the expression of different antigens within the same tumor or between primary and metastatic sites [49,50]. This heterogeneity makes it difficult for CAR-T cells, which were traditionally engineered to target a single antigen, to effectively eliminate all cancerous cells. The risk of antigen loss or mutation can result in tumor escape variants, where cancer cells that no longer express the target antigen adapt and spread, causing a relapse. Moreover, in the hostile environment of CRC, CAR-T cells often exhibit reduced persistence and activity, limiting their therapeutic potential. Therefore, it is necessary to create dual-targeting or logic-gated CAR designs. Clinical trials using CEA-targeted CAR-T cells have demonstrated promising disease stabilization, although the risk of on-target, off-tumor toxicity remains due to CEA expression in normal gastrointestinal epithelium [51]. Accordingly, multi-antigen targeting approaches, including tandem CARs or AND-gated systems (e.g., CEA + HLA-A02 LOH), may offer improved tumor specificity and reduced toxicity risk [52].

Response 4: (Answer for point.4)

About HLA expression, we totally agree with your point.

Indeed, the EMR8-5 antibody recognizes only the heavy chain of HLA class I and does not detect the trimeric complex consisting of the antigenic peptide, β2-microglobulin, and the heavy chain. However; mutations in TAP1 and TAP2, which play crucial roles in the transfer of antigenic peptides from the cytoplasm to the endoplasmic reticulum, have been reported to result in decreased surface expression of HLA class I (PMID: 10074495). Similarly, loss of the β2-microglobulin gene is known to reduce HLA class I expression (PMID: 9637706). Taking these findings into consideration, we believe that the HLA class I molecules detected on the cell surface using the EMR8-5 antibody are likely to be in a functional trimeric form composed of the heavy chain, β2-microglobulin, and an antigenic peptide

We have reported the correlation between the positivity by IHC using EMR 8-5 and the function ability of HLA using HCC tissues in 2019 [70] . In this report, we found that HCC tissues showed higher staining intensity than that in normal liver tissue by IHC using EMR 8-5, and this trend was similar to FACS analysis using W6/32 when HCC cells and normal liver cells were harvested from surgical resected HCC tissues. Moreover, accompanied by higher HLA expression, HCC cells induced more efficient activation of antigen specific CTLs. These results suggested that EMR 8-5 antibody could detect native form of HLA-ABC in FFPE tissues, and estimate whether cells are recognized by CTLs

To clarify your questions, We added in our manuscript as below.

(Page 9 lines 292-299)

Meanwhile, in terms of HLA class I, our studies were used EMR8-5, a novel pan-HLA monoclonal antibody, is suitable for HLA class I immunostaining in formalin-fixed paraffin-embedded tissue specimens. The EMR8-5 antibody recognizes all HLA-A, -B and -C heavy chains, even in formalin-fixed tissues, and these results could estimate whether cells are recognized by CTLs [70].

[70]. Akazawa Y; Nobuoka D.; Takahashi M.; Yoshikawa T.; Shimomura M.; Mizuno S.; Fujiwara T.; Nakamoto Y.; Nakatsura T. Higher human lymphocyte antigen class I expression in early‐stage cancer cells leads to high sensitivity for cytotoxic T lymphocytes. Cancer Sci 2019, 110, 1842-1852. doi:10.1111/cas.14022.

Response 5: (Answer for point.5)

Besides, referred to report of Michelakos T, et al. about the deficiency of HLA-ABC in CRC and the effect on prognosis, they performed comprehensive analysis with 5 studies from total 2863 CRC patients, and concluded that the percentage of CRC with total HLA (HLA-ABC) deficiency was 16%, and that 84% of CRC expressed either HLA-A, B, or C [71]. We believe that there are not significant differences between our results with primary CRC and their report.

In addition, on metastatic liver cancer, we previously assessed the imbalance of heterozygous on HLA gene by WES (whole exome sequencing) with total 22 metastatic liver cancer tissues from CRC origin to estimate the LOH (loss of heterozygous) of HLA gene locus (in house data, the figure A shown below). As a result, 31.8% of patients who have metastatic liver cancer of CRC origin showed the putative loss of heterozygosity of HLA gene (the figure B shown below).

 Compared with immunohistochemically analysis with RPM8-5, we found that some of cases with LOH of HLA-A, B or C showed the expression of HLA-ABC, since the staining with RPM8-5 could not distinguish clearly the cases with LOH in only one HLA gene locus, or the loss of one of HLA-A, B, or C (the figure C shown below).

For these reasons, the percentage of cases with the lower or deficient expression of HLA-ABC estimated by immunohistochemistry is always underestimated, compared to that by genomic analysis.

We added in our manuscript as below.

(Page 9 lines 307-312)

Comparable results have been reported by Michelakos T, et al in 2021 [71]. About the deficiency of HLA-ABC in CRC, they performed comprehensive analysis with 5 large cohort study of CRC and reported that the percentage of CRC with total HLA (HLA-ABC) deficiency was 16%, and that 84% of CRC expressed either HLA-A, B, or C. Using whole exome sequencing (WES) analysis, Li C et.al. also reported the positive incidence of LOH in HLA locus in primary CRC and CRCLM [72]. Although there was no significance, the frequency of LOH-HLA in CRCLM showed tendency increase compared with that in primary: the positive incidence of LOH HLA was 26% (4/15) in all patients, 13% (2/15) in primary CRC, 26% (4/15) in liver metastases (CRCLM), respectively [ref]. Meanwhile, in our house data, 31.8% of patients with CRCLM showed the putative loss of heterozygosity of HLA gene. Compared with immunohistochemically analysis with RPM8-5, we found that some of cases with LOH of HLA-A, B or C showed the positive staining of HLA-ABC, since the staining with RPM8-5 could not distinguish clearly the cases with LOH in only one HLA gene locus, or the loss of one of HLA-A, B, or C, depend on its staining intensity.

[71] Michelakos T, Kontos F, Kurokawa T, Cai L, Sadagopan A, Krijgsman D, Weichert W, Durrant LG. Kuppen PJ, Ferrone CR, and Ferrone S. Differential role of HLA-A and HLA-B, C expression levels as prognostic markers in colon and rectal cancer: Journal of Immunotherapy of Cancer 2022;10: e004115. doi:10.1136/jitc-2021-004115

[72] Li C, Xu J, Wang X, Zhang C, Yu Z, Liu J, Tai Z, Luo Z, Yi X, and Zhong Z. Whole exome and transcriptome sequencing reveal clonal evolution and exhibit immune-related features in metastatic colorectal tumors. Cell Death Dis 2021, 7,222.

Finally, we believe that our studies could be first step in immunotherapy targeting a protein substance called common cancer antigens, while paying attention to absence or presence of HLA class I expression on cancer cell membranes.

4. Response to Comments on the Quality of English Language

Point 1:  The English could be improved to more clearly express the research.

Response 1:    (in red)
These revises has been thoroughly checked, including the spelling.

5. Additional clarifications

Reviewer 2 Report

Comments and Suggestions for Authors

The authors used immunohistochemical staining of surgical specimens from CRC and liver metastases to examine the expression levels and localization of seven common cancer antigens and HLA class I. They concluded that CRC patients with high HLA class I expression may be suitable for vaccine-based and TCR-T cell therapy targeting LCDN1, EphB4, LAT1, FOXM1, HSP105a, ROBO1, and SPARC and may be suitable for vaccine-based and TCR-T cell therapy targeting these antigens, with the exception of FOXM1 and SPARC, for liver metastases.

This is an extension of the authors' study (Ref. 23), which identified 68 HLA-A24 and A2-restricted cytotoxic lymphocyte-inducing peptides derived from 10 common cancer-specific antigens frequently expressed in various solid tumors. This study attempted to apply these TAAs to the treatment of CRC and liver metastases, but there are several points that need to be reconsidered.

First, most immunohistochemical staining results are out of focus and fail to provide morphological characterization of cancer cells and surrounding normal cells (Figures 1, 3, and 4). This must be improved.

In the body of the paper, it is unclear whether it is the figure legend or the text. page 4, lines 122-129 and page 5, lines 130-131 appear to be the legend for Figure 1. page 7, lines 176-188 appear to be the legend for Figure 2. page 8, lines 208-212 and page 9, lines 213-225 appear to be the legend for Figure 3. Page 10, lines 229-238 appear to be the legend for Figure 4. If they are legends, the font should be smaller than the text.

Materials and Methods, page 12, lines 361-370: The authors have described the results shown in Figure 1 in Materials and Methods. This should be moved to the Results section.

In this study, the authors focused on gene expression in cancer and surrounding normal tissue. However, as noted on page 12, lines 368-370, in normal tissues, CLDN1 was expressed in pancreatic acinar cells and skin, LAT1 in the basal layer of pharyngeal and esophageal squamous epithelium, EphB4 in vascular endothelial cells, and HSP105a in hepatocytes. In clinical practice, the expression of these genes should be taken into account to prevent off-target effects of peptide vaccines, CAR-T cell therapy, and TCR-T cell therapy. Furthermore, the expression of these genes in the brain and heart is of greater clinical importance. Results in these organs would be useful in planning clinical trials. It would be advisable to present the results of immunohistochemical staining in a manner that does not overlap with the figures in Reference 23. Alternatively, information on gene expression in each organ could be obtained using published databases. This can be included in the results and discussion.

Titles in Tables 1 and 2: Expression of seven common cancer antigens and HLA class I on the cell membrane in 28 primary colorectal cancer cases and 60 colorectal cancer liver metastases. Gene expression was examined only on the cell membrane?

Figure 2: HLA class I expression is found at the cell membrane and in the cytoplasm. Is the expression score a combination of both? This point needs to be clearly stated.

Page 12, lines 352-356: Is there a total score from 0 to 9? This needs to be described. 

Author Response

For research article

Identification of common cancer antigens useful for specific immunotherapies to colorectal cancer and liver metastases

Response to Reviewer 2 Comments

1. Summary

Thank you very much for reviewing and commenting on our manuscript. We have revised our manuscript in accordance with your comments and suggestions. The revised parts are shown in red font in the revised version of the manuscript. Our point-by-point responses to the reviewer`s comments are given below, and we have made every effort to address the issues raised and respond to all the comments. We hope that our revisions will meet the reviewers’ expectations. Kindly refer to the page and line numbers in the revised version of the manuscript.

2. Questions for General Evaluation

Reviewer’s Evaluation

Response and Revisions

Does the introduction provide sufficient background and include all relevant references?

Can be improved

Are all the cited references relevant to the research?

Can be improved

Is the research design appropriate?

Can be improved

Are the methods adequately described?

Can be improved

Are the results clearly presented?

Can be improved

Are the conclusions supported by the results?

Can be improved

3. Point-by-point response to Comments and Suggestions for Authors

Comments 1:

The authors used immunohistochemical staining of surgical specimens from CRC and liver metastases to examine the expression levels and localization of seven common cancer antigens and HLA class I. They concluded that CRC patients with high HLA class I expression may be suitable for vaccine-based and TCR-T cell therapy targeting LCDN1, EphB4, LAT1, FOXM1, HSP105a, ROBO1, and SPARC and may be suitable for vaccine-based and TCR-T cell therapy targeting these antigens, with the exception of FOXM1 and SPARC, for liver metastases.

This is an extension of the authors' study (Ref. 23), which identified 68 HLA-A24 and A2-restricted cytotoxic lymphocyte-inducing peptides derived from 10 common cancer-specific antigens frequently expressed in various solid tumors. This study attempted to apply these TAAs to the treatment of CRC and liver metastases, but there are several points that need to be reconsidered.

First, most immunohistochemical staining results are out of focus and fail to provide morphological characterization of cancer cells and surrounding normal cells (Figures 1, 3, and 4). This must be improved.

Response 1:

Thank you very much for reviewing and commenting on our manuscript.

We revised the quality of figure more clearly according to your suggestion. Please confirm these figures.(Page 5,11,13)

Comments 2:

In the body of the paper, it is unclear whether it is the figure legend or the text. page 4, lines 122-129 and page 5, lines 130-131 appear to be the legend for Figure 1. page 7, lines 176-188 appear to be the legend for Figure 2. page 8, lines 208-212 and page 9, lines 213-225 appear to be the legend for Figure 3. Page 10, lines 229-238 appear to be the legend for Figure 4. If they are legends, the font should be smaller than the text.

Response 2:

Thank you very much for reviewing and commenting on our manuscript. As you pointed out, they were all legends, so we made the text smaller and reflected that in our revision.

Comments 3:

Materials and Methods, page 12, lines 361-370: The authors have described the results shown in Figure 1 in Materials and Methods. This should be moved to the Results section.

Response 3:

Thank you very much for reviewing and commenting on our manuscript.

We moved previous sentence to the Results section according to your suggestion.

(page2 line 101-110)

At first, Figure 1 shows the immunostaining patterns and scoring positive staining for each common cancer antigen and HLA class I. SPARC was additionally expressed in the cytoplasm of cancer-associated fibroblasts (CAFs) in addition to cancer cell cytoplasm, suggesting potential therapeutic relevance for targeting SPARC in both cancer cells and CAFs [47-51] Although ROBO1 is a transmembrane protein receptor, its expression is predominantly cytoplasmic in primary CRC and liver metastases [52, 53]. Therefore, we assessed ROBO1 expression in the cytoplasm. In normal tissues, CLDN1 was expressed in pancreatic acinar cells and skin, LAT1 in the basal layer of pharyngeal and esophageal squamous epithelium, EphB4 in vascular endothelial cells, and HSP105α in hepatocytes [54].

Comments 4:

In this study, the authors focused on gene expression in cancer and surrounding normal tissue. However, as noted on page 12, lines 368-370, in normal tissues, CLDN1 was expressed in pancreatic acinar cells and skin, LAT1 in the basal layer of pharyngeal and esophageal squamous epithelium, EphB4 in vascular endothelial cells, and HSP105a in hepatocytes. In clinical practice, the expression of these genes should be taken into account to prevent off-target effects of peptide vaccines, CAR-T cell therapy, and TCR-T cell therapy. Furthermore, the expression of these genes in the brain and heart is of greater clinical importance. Results in these organs would be useful in planning clinical trials. It would be advisable to present the results of immunohistochemical staining in a manner that does not overlap with the figures in Reference 23. Alternatively, information on gene expression in each organ could be obtained using published databases. This can be included in the results and discussion.

Response 4:

Thank you very much for reviewing and commenting on our manuscript.
Our division investigated the expression of these seven common cancer antigens in normal tissues [54].

l  ROBO1 staining was observed in the basal layer cells of the squamous epithelium of the pharynx and esophagus and in the mucosal epithelium cells of the stomach and gallbladder.

l  EphB4 staining was observed in the basal layer cells of the squamous epithelium of the pharynx and esophagus and in the mucosal epithelial cells of the stomach, gallbladder, and large intestine.

l  CLDN1 staining was observed in the acinar cells of the pancreas and skin.

l  LAT1 staining was observed in the cells of the basal layer of the squamous epithelium of the pharynx and esophagus, which was relatively strong.

l  FOXM1 and SPARC staining were observed in the digestive tract.

Although these antigens showed the staining in some normal tissues, it was always moderate and weaker compared with that in cancerous area. Moreover, the expression pattern of HLA class I in normal tissues was mostly cytoplasmic or has no expression, probably to avoid the autoimmunity by CTLs. In particular, the staining of HLA class I on the cell membrane was observed in the basal layer cells of the squamous epithelium of the pharynx and esophagus, mucosal epithelial cells of the digestive tract, and stratified squamous epithelium of the skin.  Therefore, careful progress would be necessary, when administering vaccines or TCR-T with antigens that are expressed in these cells.

[54]. Nakatsura T.; Takenouchi, K.; Kataoka J.; Ito Y.; Kikuchi S.; Kinoshita H.; Ohnuki K.; Suzuki T.; Tsukamoto N.; Expression Profiles of Five Common Cancer Membrane Protein Antigens Collected for the Development of Cocktail CAR-T Cell Therapies Applicable to Most Solid Cancer Patients. Int J Mol Sci 2025, 26(5), 2145. doi:10.3390/ijms26052145.

In our manuscript, we analyzed CRC and CRCLM separately. Furthermore, we analyzed subgroups based on whether native or chemotherapy was administered. There is no overlap with Reference 23

Comments 5:

Titles in Tables 1 and 2: Expression of seven common cancer antigens and HLA class I on the cell membrane in 28 primary colorectal cancer cases and 60 colorectal cancer liver metastases. Gene expression was examined only on the cell membrane?

Response 5:

Thank you very much for reviewing and commenting on our manuscript.

For HLA class I expression, the presence and localization (cell membrane or cytoplasm) of staining were evaluated in cancer cells, while our studies were defined that expression on the cell membrane of cancer cells was only positive.

In terms of expression of seven common cancer antigens, the Membrane proteins, such as CLDN1, EphB4, and LAT1 were expressed in cancer cell membrane. Other intracellular molecules, such as FOXM1, HSP105α, ROBO1, and SPARC were expressed on cancer cell nuclei, cytoplasm of cancer cell, and cytoplasm of cancer-associated fibroblasts (CAF).

Comments 6:

Figure 2: HLA class I expression is found at the cell membrane and in the cytoplasm. Is the expression score a combination of both? This point needs to be clearly stated.

Response 6:

Thank you very much for reviewing and commenting on our manuscript.

As with other common cancer antigens, the score combination is the sum of the staining intensity and positive area of cancer cells. For HLA class I, only the expression in the cell membrane is calculated as staining intensity positive. These points will be stated and added in our manuscript.

(Page 6 lines 181-183, Figure 1 legend)

For the scoring of HLA class I expression, the distinct staining on the edge of each tumor cells was defined as positive staining, and was evaluated the expression of HLA.

Comments 7:

Page 12, lines 352-356: Is there a total score from 0 to 9? This needs to be described.

Response 7:

Thank you very much for reviewing and commenting on our manuscript.

The immunostaining total scores for each common cancer antigen and HLA class I are the sum of staining intensity (negative: 0, weakly positive: 1, weakly to strongly positive: 2, and strongly positive: 3) and positive area (The percentages of positively stained cells areas were scored as follows: 0 (0–10%), 1 (10–39%), 2 (40–69%), and 3 (70–100%).), so the score ranges from 0 to 6. This point will be stated and added in ”material and method” on our manuscript as below.

(Page 14, line 428-434)

Staining intensity of the tumor tissue was scored for each of the seven antigens using a semi-quantitative method (negative: 0, weakly positive: 1, weakly to strongly positive: 2, and strongly positive: 3), while the percentages of positively stained cells areas were scored as follows: 0 (0–10%), 1 (10–39%), 2 (40–69%), and 3 (70–100%). We defined the sum of these two scores as the total score.

Total score (0~6)  =

 Average scores of staining intensity from three regions of interest (ROIs): 0~3 

+  Average scores of positive areas from three ROIs: 0~3

5. Additional clarifications

[Here, mention any other clarifications you would like to provide to the journal editor/reviewer.]

Reviewer 3 Report

Comments and Suggestions for Authors

I enjoyed reading this paper which highlights some intriguing targets for anti-colorectal cancer immunotherapies. Some of these may also function as "pan-cancer" antigens, so any attention shed on them is worthwhile. 

Author Response

Comments 1 :
I enjoyed reading this paper which highlights some intriguing targets for anti-colorectal cancer immunotherapies. Some of these may also function as "pan-cancer" antigens, so any attention shed on them is worthwhile
Response 1: Thank you very much for reviewing and commenting of our manuscript.

Round 2

Reviewer 1 Report

Comments and Suggestions for Authors

Most of the comments and suggestions have been addressed by the authors in the revised version of the manuscript. However, new errors have emerged. For example, belowe the Figure 1 there is a statement: "Loss of heterozygosity was defined as a lack of expression in the cytoplasm or cell membrane." This is a serious error—loss of heterozygosity (LOH) cannot be determined by immunohistochemical analysis. This study does not assess loss of heterozygosity (LOH) affecting HLA class I genes. Additionally, there is an error on page 7: "Comparable results have been reported by Michelakos T, et al in 2021 [71]." The correct publication year is 2022. 

Missing ference number in; "Although there was no significance, the frequency of LOH-HLA in CRCLM showed tendency increase compared with that in primary: the positive incidence of LOH HLA was 26% (4/15) in all patients, 13% (2/15) in primary CRC, 26% (4/15) in liver metastases (CRCLM), respectively [ref]."

Author Response

Response to Reviewer 1 Comments

1. Summary

Thank you very much for reviewing and commenting on our manuscript. We have revised our manuscript in accordance with your comments and suggestions. The revised parts are shown in red font in the revised version of the manuscript. Our point-by-point responses to the reviewer`s comments are given below, and we have made every effort to address the issues raised and respond to all the comments. We hope that our revisions will meet the reviewers’ expectations. Kindly refer to the page and line numbers in the revised version of the manuscript.

2. Questions for General Evaluation

Reviewer’s Evaluation

Response and Revisions

Does the introduction provide sufficient background and include all relevant references?

Yes

Is the research design appropriate?

Yes

Are the methods adequately described?

Yes

Are the results clearly presented?

Can be improved

Are the conclusions supported by the results?

Yes

Are all figures and tables clear and well-presented?

Can be improved

3. Point-by-point response to Comments and Suggestions for Authors

Comments 1:

Most of the comments and suggestions have been addressed by the authors in the revised version of the manuscript. However, new errors have emerged. For example, below the Figure 1 there is a statement: "Loss of heterozygosity was defined as a lack of expression in the cytoplasm or cell membrane." This is a serious error—loss of heterozygosity (LOH) cannot be determined by immunohistochemical analysis. This study does not assess loss of heterozygosity (LOH) affecting HLA class I genes.

Response 1:

Thank you very much for reviewing and commenting on our manuscript. Regarding the definition of loss of heterozygosity (LOH), your suggestions are correct. According to your suggestions, we will delete the sentence below (Figure 1 Legend Page 5 Line 145-146).

“Loss of heterozygosity was defined as a lack of expression in the cytoplasm or cell membrane.”

In addition, our manuscripts are revised on your point, and changed the sentence, “and 3 cases (5.00%) of loss of heterozygosity” (Results section, Page3, Line 129-131), as below,

“In three cases (5.00%), HLA expression was not detected on both cytoplasm and cell membrane.”

Comments 2:

Additionally, there is an error on page 7: "Comparable results have been reported by Michelakos T, et al in 2021 [71]." The correct publication year is 2022

Response 2:

Thank you very much for reviewing and commenting on our manuscript. We are sorry to make such as mistake. As you pointed out, We revised this sentence.

Comments 3:

Missing reference number in; "Although there was no significance, the frequency of LOH-HLA in CRCLM showed tendency increase compared with that in primary: the positive incidence of LOH HLA was 26% (4/15) in all patients, 13% (2/15) in primary CRC, 26% (4/15) in liver metastases (CRCLM), respectively [ref]." .

Response 3:

Thank you very much for reviewing and commenting on our manuscript.

These sentences was referred by Li C et al [72], so we add [72] at the end of these sentences..

5. Additional clarifications

[Here, mention any other clarifications you would like to provide to the journal editor/reviewer.]

Reviewer 2 Report

Comments and Suggestions for Authors

The authors respond appropriately to the points raised by the reviewer.

Author Response

For research article

Identification of common cancer antigens useful for specific immunotherapies to colorectal cancer and liver metastases

Response to Reviewer 2 Comments

1. Summary

Thank you very much for taking the time to review this manuscript.

2. Questions for General Evaluation

Reviewer’s Evaluation

Response and Revisions

Does the introduction provide sufficient background and include all relevant references?

Yes

Are all the cited references relevant to the research?

Yes

Is the research design appropriate?

Yes

Are the methods adequately described?

Yes

Are the results clearly presented?

Yes

Are the conclusions supported by the results?

Yes

3. Point-by-point response to Comments and Suggestions for Authors

Comments 1:

The authors respond appropriately to the points raised by the reviewer.

Response 1: Thank you very much for reviewing and commenting of our manuscript.
